# Sustainable Hydrothermal and Solvothermal Synthesis of Advanced Carbon Materials in Multidimensional Applications: A Review

**DOI:** 10.3390/ma14175094

**Published:** 2021-09-06

**Authors:** Lwazi Ndlwana, Naledi Raleie, Kgogobi M. Dimpe, Hezron F. Ogutu, Ekemena O. Oseghe, Mxolisi M. Motsa, Titus A.M. Msagati, Bhekie B. Mamba

**Affiliations:** 1Florida Science Campus Florida, Institute for Nanotechnology and Water Sustainability (iNanoWS), College of Science, Engineering and Technology, University of South Africa, Johannesburg 1709, South Africa; ralein@unisa.ac.za (N.R.); ogutuho@unisa.ac.za (H.F.O.); osegheo@unisa.ac.za (E.O.O.); motsamm@unisa.ac.za (M.M.M.); msagatam@unisa.ac.za (T.A.M.M.); mambabb@unisa.ac.za (B.B.M.); 2Doornfontein Campus, Department of Applied Chemistry, University of Johannesburg, P.O. Box 17011, Johannesburg 2028, South Africa; mdimpe@uj.ac.za; 3School of Materials Science and Engineering, Tianjin Polytechnic University, Tianjin 300387, China

**Keywords:** carbon materials synthesis and functionalisation, energy, graphene, gas separation, hydrothermal and solvothermal carbonisation, microwave-assisted synthesis, nanocomposite membranes, sensing, water treatment

## Abstract

The adoption of green technology is very important to protect the environment and thus there is a need for improving the existing methods for the fabrication of carbon materials. As such, this work proposes to discuss, interrogate, and propose viable hydrothermal, solvothermal, and other advanced carbon materials synthesis methods. The synthesis approaches for advanced carbon materials to be interrogated will include the synthesis of carbon dots, carbon nanotubes, nitrogen/titania-doped carbons, graphene quantum dots, and their nanocomposites with solid/polymeric/metal oxide supports. This will be performed with a particular focus on microwave-assisted solvothermal and hydrothermal synthesis due to their favourable properties such as rapidity, low cost, and being green/environmentally friendly. These methods are regarded as important for the current and future synthesis and modification of advanced carbon materials for application in energy, gas separation, sensing, and water treatment. Simultaneously, the work will take cognisance of methods reducing the fabrication costs and environmental impact while enhancing the properties as a direct result of the synthesis methods. As a direct result, the expectation is to impart a significant contribution to the scientific body of work regarding the improvement of the said fabrication methods.

## 1. Introduction

Carbon materials are extremely versatile in terms of their applications, and as such, there is a growing global demand for these materials. This demand, in turn, necessitates the development of rapid, cost-effective, scalable, and safe synthesis methods. These synthesis methods include electric arc-discharge, ionothermal, pulse laser ablation, chemical vapour deposition (CVD), and the pyrolysis of organic compounds/solvents, among other approaches [1,2,3,4]. However, these methods are known to be energy intensive, they require harsh conditions and chemicals, and they are quite expensive to carry out [5]. Consequently, a shift towards more favourable synthesis methods has been observed because most of the aforementioned conventional methods possess several drawbacks such as product contamination, lengthy reaction times, the use of catalysts and toxic chemicals, among others [6]. To achieve sustainable carbon materials with enhanced properties for wide applicability, approaches such as sol–gel, hydrothermal, and solvothermal syntheses which with several modifications and changes have shown great promise over the recent past [7]. The parameters allowing this are solely dependent on the reaction conditions such as the precursor type/concentrations, temperature, pressure, and reaction time [4]. The uses of these carbon materials have found inclusion in energy, environmental/biological sensing, water treatment, and gas purification/sensing [2,8,9].

There has been considerable interest in the development of carbon materials in the recent past. In the description of the methods mentioned above, the first is the sol–gel method of synthesis [8]. This method usually involves the preparation of catalysts to serve the carbonisation (thermal treatment) of different precursors for the carbon nanomaterials to be utilised in various applications such as capacitive deionisation (desalination) of saline water, environmental sensing, gas sensing/separation, and energy production, among others [1,4,9,10,11,12]. As such, the tunable morphologies and chemical compositions of advanced carbon materials (and their modified counterparts) have been prepared [13].

Such morphologies of carbon materials have been achieved through the advancements in solvothermal synthesis which involves the use of solvents and high temperatures during the preparation of the carbon materials. In one study, a facile solvothermal carbonization of organic components for the synthesis of Fe_3_O_4_@C as an anode material for energy applications (lithium ion batteries) was presented [14]. Similar cyclable metal oxide/C anode materials have been prepared, such as the α-MoO_3_/MWCNT, prepared via a surfactant-assisted solvothermal method followed by low-temperature calcination [15]. In this regard, low-temperature calcination already implies reduced working costs and more sustainable processes. Other works towards the synthesis of battery materials possessing high discharge capacities has also been published [16,17]. Among the various challenges associated with solvothermal synthesis include the high demand for energy because of the high temperatures required to satisfy the carbonisation. On the other hand, the hydrothermal synthesis of materials will have the reactants or carbon source dispersed in simple clean water as opposed to a solvent prior to the carbonisation under the conditions of high temperatures and pressures. The benefits of this method include the ease of synthesis and of composition control, and thus regulating the resulting particle properties such as high surface area, size, shape, and dispersion [18,19]. The advantages of these synthesis approaches of, also include thermal and chemical stability which act in synergy to mitigate dissolution and leaching when metals are implemented as dopants to the carbon material used [20]. As previously mentioned, one of the caveats of these methods lies in energy demand due to the high reaction temperatures. As such, other methods are being developed to reduce energy demand. These methods are low-temperature hydrothermal or solvothermal approaches.

Low-temperature solvothermal synthesis (LTS) involves the use of milder conditions to fabricate a required material in the presence of an organic solvent. This mostly prefers solvents for the carbonisation of precursors. LTS is not actually a new method because as far back as 2004, Kuang et al. (2004) presented the preparation of crumpled carbon nanosheets (CCNs) in the presence of CCl_4_, which is widely preferred for the synthesis of CCNs. The resulting structure was of a crystalline nature with a thickness of several nanometres and flowerlike morphology. The superior advantage of this method is the use of no catalysts and that it occurs under extremely low heating, indicating low energy demand. The main disadvantage of this route is the use of the toxic solvent, CCl_4_ [21]. Recently, Manohara et al. (2019) prepared an aluminium-functionalised carbon-based nanocomposite using LTS for a carbon membrane, successfully removing several environmental insults from water [9]. To further address challenges such as complicated procedures/equipment and environmental threats posed by high-temperature approaches, an even simpler method to prepare graphene nanosheets was presented by Ye et al. (2020). This method involved a one-pot reaction without the use of harsh chemicals [22]. There are also two low-temperature approaches called hydrothermal and solvothermal carbonisation (HTC and STC, respectively). These reactions take place under mild conditions to result in coal-like products called hydrochars or biochars. Hydrochars are products of hydrated carbon-based materials such as sugars and other organic materials, organic compounds (solvents or polymers). Biochars, however, are produced from biomass/biowastes such as grass, nutshells, wood, and cellulosic bagasse, among others [23]. Feedstocks such as wood and nutshells produce highly ordered (hierarchical) porous structures whereas polymers, solvents, and bagasse do not. Hierarchically structured materials possess high porosities and surface areas owing to their different modalities of pore length structures within them. These pore lengths can vary from macro (<50 nm), meso (2–50 nm), down to micro (<2 nm) [24]. However, with incoming developments to these synthesis routes, ordered nanostructures from different feedstocks are now possible through CVD, templating, spray pyrolysis, and HTC [2].

With these developments in hydrothermal and solvothermal approaches, the methods are still slow with unsatisfactory yields. As such, MWA-STC and MWA-HTC have gained great interest due to its rapid reactions, cleaner products, and the decreased use of solvents and catalysts (for MWA-STC). Additional benefits to MWAS are the improved yields and lower power consumption, thereby reducing costs. For MWA-HTC, the reactions can even occur without the use of catalysts or harsh/toxic chemicals, thereby promoting green chemistry [25,26]. The benefit of microwave irradiation lies in the mechanism of heating, which is more efficient than conventional heating used by traditional STC and HTC. As reviewed by Remya and Lin (2011), the principal heating mechanisms in MWAS are dipolar polarization (DP, which most MWA heating mechanisms follow), conduction mechanism (CM), and interfacial polarization (IP). In DP, the MW radiation interacts with the dipole of a material at a molecular level and initiates dipole rotation. Dipole rotation, in turn, causes a dipole delay and increases the energy of the system, and this results in heating. In CM, the electrical conductors interact with the MW irradiation and cause the rapid movement of charge carriers like ions, electrons, etc. in the entire material and that results in heating. The combination of these two mechanisms then results in IP. This mechanism is of great importance when heating both conducting and insulating materials [27]. Due to the advantages of the principal heating mechanism in MWAS, it has become prominent in STC and HTC for the synthesis of advanced carbon materials. These can be microporous, non-ordered, hierarchical carbon materials, and their resulting nanocomposites. The properties of these carbon materials can be easily tailored by adjusting the magnetron power, reaction time, temperature, and reactant concentrations. All these parameters can manipulate the resulting morphologies, porosity, and particle size (among others) and thus the properties of the material. The properties, in turn, make these materials suitable for energy, water treatment, membrane nanocomposite technology, gas sensing/purification, and environmental/biosensing applications.

Although over the past six years, for the most part, there have been many reviews that have been written about the synthesis of carbon materials, there is a lack of literature on the recent developments of sustainable and green/er synthesis methods and their applications thereof. This review, as such, discusses favourable, rapid, and environmentally friendly fabrication methods and the enhanced properties they impart on the resulting advanced carbon materials. LTLP and LTHP synthesis methods will also be interrogated. A particular emphasis will be given to microwave chemistry approaches, which also lacks reviews when compared to other methods. An overview of the use of sustainable carbon sources such as biomass, biowastes, waste solvents, waste polymers, and other carbon precursors will also be provided in this review. Furthermore, the synthesis conditions and advantages/disadvantages of these methods will be disseminated. The resulting properties such as the particle size, shape, and morphology, among others, will also be discussed. An outlook will also be given to inspire further exiting offerings in the future.

## 2. Short Summary of the Development of Advanced Carbon Materials Synthesis Methods and Applications

The development and growth of the solvothermal/hydrothermal methods are directly related to the creation of nanomaterials. The first study on the solvothermal process can be traced back to the middle of the nineteenth century when micrometre-to-nanometre-sized quartz particles were used to prepare these carbon materials. However, between the 1840s and 1990s, the development and introduction of the hydrothermal method in material synthesis lagged behind, due to the unavailability of the instrumentation for nanomaterial characterisation [28]. Furthermore, at that time, only a limited understanding of the solution chemistry was available to effectively monitor crystal growth [18]. In the 1990s, hydrothermal methods were resurrected along with the revolution in nanomaterials and the advent of high-resolution microscopes from the 1980s. At the same time, significant improvements were made to comprehend the physicochemical features of hydrothermal processes, leading to the invention of the solvothermal synthesis method where organic compounds were used as solvents towards the preparation of well-controlled nanoscale materials [28]. Due to these superior advantages, including low process temperatures, reaction productivity in liquid media, low-energy consumption, and being environmentally friendly, the hydrothermal/solvothermal processes has been made in the twenty-first century in developing nanomaterials with the control of physicochemical properties such as crystallinity, crystal phase, morphology, and size [10,28].

Ever since the discovery of fullerenes in 1985 by Kroto and co-workers, just one year before the discovery of carbon soot, which was synthesised for the first time in an inert atmosphere from graphite by a resistive heating process [29]. Many new allotropes of carbon have since been discovered. Among these are carbon nanotubes (CNTs), which have received considerable attention since the work by Iijima in 1991 [30], and graphene, which is a single layer of carbon atoms arranged in a honeycomb lattice, discovered in the year 2004 by Novoselov and workers [31]. Graphene is formed out of a flat monolayer of carbon atoms which are densely packed in a two-dimensional honeycomb lattice. The carbon atoms are sp^2^ hybridised; hence, they form strong intra-layer bonding within the hexagonal carbon rings, which makes graphene one of the strongest materials to be prepared [32]. This results in carbon materials finding a myriad of applications where developments have led to the discovery of many applications for carbon. Its allotropes and many structures/modified permutations have been applied in water treatment, gas separation, energy, photovoltaics, and the other aforementioned applications [33,34]. These include the use of carbon nanomaterials in the catalytic detection, removal, and degradation of organic compounds in water through photocatalysis, electrocatalysis, sonocatalysis, and all their respective amalgamations [35,36]. Furthermore, these approaches have been included in the development of mixed-matrix membranes for water treatment from microfiltration to reverse osmosis (RO) applications.

In terms of the synthesis and modification methods, major approaches include hydrothermal and solvothermal techniques, with the green/er synthesis methods being among those recently gaining considerable interest. The use of toxic and expensive solvents is losing favour because these are difficult and costly to remove. Among such new approaches are hydrothermal and solid phase thermal methods.

## 3. Advanced Synthesis Methods and the Modification of Carbon Materials

In this section, a general overview of the different synthesis methods of carbon and carbon-based materials is given. The important role of the synthesis parameters that were demonstrated to be key factors during the growth process is also outlined. The results summarised in this work are essentially the studies that were conducted largely during the past five years.

### 3.1. The Arc-Discharge Method

The arc-discharge method was first used by Krätschmerand Hoffman in 1990 to synthesise fullerene (C_60_) [37]. This technique generally involves the use of two high-purity graphite electrodes, i.e., anode and cathode. The cathode and anode are both pure graphite rods, and the current in the discharge process is maintained at 100–150 A. The electrodes are then briefly brought into contact and an arc is struck. This method involves the use of an electric arc oven that mainly comprises two electrodes and a steel chamber cooled by water. Li et al. (2010) synthesised N-doped multi-layered graphene in the mixing atmosphere of He and NH_3_. The graphene, as synthesised through this method, possessed distinct characteristics and can be utilised in different applications such as electronics, gas sensors, energy storage, and water treatment, among others [38].

In 2015, Sharma and co-workers synthesised multi-walled carbon nanotubes (MWCNTs) using the arc-discharge method, where an arc was produced in between the electrodes by a D.C. power supply capable of providing 100–200 Amps of current, and a voltage range of 20–30 V. The reaction environment was maintained by an open vessel containing deionised water, CNTs having a lesser number of layers and diameter in the range 15–150 nm were obtained. The arc-discharge-synthesised MWCNTs were treated with eight M HNO_3_ and eight M H_2_SO_4_ solutions and it was revealed by XRD that CNTs synthesised by arc-discharge possessed good crystallinity with low amounts of impurities. The MWCNTs grown using this method are short, thick, and curved, as seen in Figure 1. Good yields can also be attained using this synthesis route with these MWCNTs finding implementation in pharmaceutical applications [39].

Pham et al. (2019) presented a simple arc-discharge approach to the doping of graphene with nitrogen [40]. This facile method involved the use of anodic C fillers, enhancing N-doping by 2–3-fold as compared to N-graphene prepared via conventional arc-discharge. This is, nonetheless, while maintaining high crystallinity and increasing the number of exposed graphene edges. As a direct consequence, this advanced carbon material indicated improved areal capacitance and pseudocapacitance. Further developments have been made such as the work by Wang et al. (2021) where the authors presented a new route for the preparation of graphene nanoflakes. The nanoflakes were prepared via non-thermal plasma and pressure-dependent process from CH_4_ and H_2_ in the presence of argon gas. This method consumed low amounts of energy (ca. 0.13 kWh/g at 200 kPa) while at variable pressures, nanoflakes with high surface areas, crystallinity, and thermal stability [41]. Such developments open a great window of potential regarding the fabrication of supercapacitors and the upscaling of graphene nanoflakes in the future.

### 3.2. Chemical Vapour Deposition (CVD)

As a synthesis method, CVD pertains to the formation of thin films on a heated substrate through a gas phase reaction. The advantage of such an approach is the ability to fine-tune the deposition rates by manipulating the reaction conditions. This can then lead to highly pure and well-structured materials with enhanced properties. However, the low-temperature CVD possesses several synthesis drawbacks such the difficulty to control the desired stoichiometry, the trapping of precursors, and a lack of product crystallinity. Pertaining to the preparation of 2D carbon films (mostly graphene), CVD is still limited to the use of Cu as a deposition substrate or its alloys. Advances have been made in conventional CVD with methods such as meta-organic CVD (MOCVD) and plasma-enhanced CVD (PECVD) gaining research traction [42]. An example of latter is a work carried out by Dissanayake and colleagues (2016) using PECVD to fabricate a spontaneous and strong multilayer of graphene with its n-doping by soda-lime glass. In this method, graphene was deposited in a non-chemical manner onto glass, achieving a highly stable nanocomposite with enhanced electron densities which increase almost two-fold upon the deposition of graphene onto a CuInGaSe_2_ (CIGS) semiconductor substrate (p-doping). The characterization of both these new materials is illustrated in Figure 2. Figure 2A shows the normalised conductance (G) vs. the gate voltage (V) of the materials where a Dirac point of −67 V was recorded, indicative of successful n-doping. Density-functional theory calculations (DFT) and its resulting curve (Figure 2B) corroborated the latter n-doping by the Fermi energy shift of +474 meV. Secondary ion mass spectrometry depth profiles for Na and Se (Figure 2C) indicate that other chemical interactions take place during the fabrication of these nanocomposites. This is given because the concentration of Na increases towards the surface from the bulk CIGS and glass. Figure 2D illustrates attests to the multilayer nature graphene on the substrate where these layers totalled a thickness of 340 pm when studied on HR-TEM. Raman spectroscopy further indicated that graphene was not destroyed during the transfer onto the rough CIGS substrate (Figure 2E). Figure 2F shows the EDS of Na and C in the respective colours of yellow and purple on the graphene and CIGS interface, suggesting interactions between the two species. Figure 2G illustrates the n-doping between the graphene and the substrate which result in this stable nanocomposite, making it suitable for supercapacitors, sensing, photovoltaics, and batteries [43]. Further improvements in the development of similar synthesis methods have led to the prevention of toxic gaseous by-products. This feat allows for further exploits for this kind of synthesis in semi-open environments [44].

### 3.3. Other Strategic Synthesis Methods of Carbon-Based Materials

#### 3.3.1. Sol–Gel Approaches for Water Treatment

In recent years, great progress has been made in the preparation of carbon and carbon-based materials. Depending on the carbon source used, the synthesis methods can generally be divided into “top–down” and “bottom–up” approaches [31]. Recently, Hlongwa and co-workers (2020) synthesised graphene through the reduction of GO prepared from the Hummer’s method. The authors reported thin, aggregated, and wrinkled graphene nanosheets. The wet chemical method was further used to synthesise graphene/Ni-doped LiMnPO_4_ (G-LMNP), and the results showed outstanding peak current intensity from electrochemical characterisation. The result was enhanced electrochemical reversibility, and this can be ascribed to the highly conductive graphene layers. These graphene layers provided new pathways for electron transfer, and thus facilitating the redox reaction [45]. The graph given on Figure 3 suggests the suitability of the G-LMNP for high-power energy storage applications.

Mamba and co-workers (2016) also used the sol–gel method to prepare tridoped titania/MWCNTs [46]. In this method, the Gd_2_O_3_ nanoparticles were introduced to the MWCNTs during the sol–gel step. This synthesis selection was then reported to enhance photocatalytic efficiency due to the improved charge separation of TiO_2_. In the previous year, Mamba et al. (2015) presented an analogous study where they interrogated the effect of the kind of CNT used (SWCNT and MWCNT) on the degradation of Eriochrome black T and eosin blue shade. The SWCNT/Nd,N,S–TiO_2_ was reported to possess higher photocatalytic efficiency than its MWCNT counterpart. This was attributed to the narrower band gap that the former indicated over the latter [47]. In addition to multidoping, other developments in manipulating reactants in sol–gel approaches have been reported. One example of such was presented by Escamilla et al. (2020). The authors eliminated water from the reaction during the precursor mixing step. Highly ordered TiO_2_/MC structures were produced, which enhanced catalytic properties during hydrogen production. The notable advantages of this method, together with many sol–gel approaches with calcination/carbonisation steps, are the lower temperatures (400 °C) used [48]. The latter feat may encourage the future adoption of such methods, further made attractive by the shorter reaction times (2 h) that sufficed the larger formation of the desired anatase phase.

#### 3.3.2. Microwave-Assisted Synthesis for Advanced Carbon Materials in Energy and Sensing

As an alternative approach to other synthesis methods, MWAS has gained considerable traction in terms of research interest that several advances were made regarding its application in solvothermal, hydrothermal, and dry syntheses. As was mentioned previously, appreciable benefits are gained through this approach. These benefits include enhanced rapidity, green/er reactions, cleaner products, reduced costs, and improved safety. A recent application of this method was the solvothermal synthesis of phloroglucinol (precursor)-based luminescent nitrogen-doped CDs by Khavlyuk et al. (2021) [49]. The authors investigated the effects of thermal conditions during the conventional and MWA solvothermal synthesis on the morphology of the resulting CD energy structure, morphology, and thus optical properties. The study suggested that the different solvents (ethylenediamine, DMF, and formamide) used during the syntheses affected the resulting material properties. These were characterised on several analytical techniques such as photoluminescence (PL), UV–vis, Raman spectroscopy/spectrometry, and HR-TEM. It is worth noting that a striking difference is the fact that the conventional solvothermal method needed extended reaction periods (6 h) to complete while the MWAS approach required only 6 min. The MWAS route also resulted in the formation of molecular moieties (or “surface states”) on the surface of the resulting CDs as detected by FTIR, PL, and XPS.

Recently, Hoang et al. (2020) synthesised GQDs via MWA-HTC for solar cell applications. They achieved these GQDs by subsequently treating the GO with ammonia hydrothermally reacting the mixture in a microwave oven at 700 W in 10 min. The GQDs obtained were between 2 and 8 nm in size with appreciable dispersion in polar solvents [26]. Chae et al. (2017) pyrolysed A2/B_3_ type monomer sets utilising MWAS-induced thermal polyamidation and carbonisation and obtained the water-soluble CQDs with QY of 23.3% and the synthesis process was completed within 3–5 min. They fabricated the CQDs from chitin nanofibres using an MWA-HTC, which took approximately 3 min. The nanocomposite was used for drug sensing based on the quenching effect. The fabricated CQDs exhibited high stability and sensitivity fluorescence to *D*-penicillamine [50].

Zirconium (Zr)-based metal–organic frameworks (MOFs) microwave-assisted synthesis was carried out by Vakili et al. (2018). The yield and porous properties of UiO-67 were optimised by modifying the modulator quantity (benzoic acid and HCl), reaction time, and temperature. They observed that an improvement in modulator quantity improved the real surface area and pore volume of UiO-67 due to the promotion of linker deficiency; and the involvement of modulators impaired the numbness of the surface area and pore volume. For synthesising UiO-67 under microwave irradiation, optimum quantities of BenAc and HCl were calculated as 40 mol equivalent and 185 mol equivalent (to Zr salt), respectively. Microwave methods have enabled faster synthesis with a reaction time of 2–2.5 h (at equivalent temperatures of 120 °C and 80 °C for BenAc and HCl, respectively) relative to traditional solvothermal synthesis, which usually takes 24 h. In microwave-assisted synthesis, the thermal influence of the microwave is assumed to lead to the rapid synthesis of UiO-67. The efficiency of reaction mass and space–time yield indicate that the simple but highly effective preparation of Zr-based MOFs was promoted by microwave heating. In addition, UiO-67 MOFs were evaluated using single component (CO_2_ and CH_4_) adsorption from various synthesis methods, i.e., microwave-assisted and solvothermal methods), showing similar gas uptakes [51]. In closed batch processes, Glover and Mu (2018) synthesised metal–organic frameworks (MOFs), however, these processes are not favourable for large-scale production. Based on a continuous flow tubular reactor fitted with microwave volumetric heating, the authors reported a scalable MOF synthesis path. Under relatively mild conditions, the reaction took only 50 min to complete. This device allowed the continuous crystallisation of MIL-100 (Fe) with a high space–time yield of ~771.6 kg/m^3^/d. The MIL-100(Fe) was used as a support for the preparation of Cu(I)-modified π complexation adsorbents. The adsorbents exhibited favoured CO adsorption over CO_2_, and the efficiency of adsorption (CO adsorption capability and CO/CO_2_ selectivity) was comparable to or even greater than most Cu(I)-modified π complexation adsorbents previously mentioned [52].

## 4. Conventional Hydrothermal Methods for the Preparation and Modification of Carbon Materials

As mentioned previously, the disadvantages of HTHP synthesis include the use of toxic reactants and the high costs incurred through the energy-intensive synthesis conditions. With this knowledge, LTLP processes have generated great interest due to lower energy demands and more environmentally friendly processes. Thus, the result is the wider applicability of the synthesis routes to develop versatile materials for large-scale industrial use.

The low-temperature-low-pressure (LTLP) approaches for the synthesis of carbon materials are termed hydrothermal carbonisation (HTC). As a method, HTC lowers working temperatures to 150–350 °C from the ~500 °C for HTHP synthesis. An advantage of this approach can also be reduced operating pressures because these are autogenerated by the reaction itself (as low as ~1 MPa). The synthesis method to result from such low working pressures is then referred to as low temperature and low-pressure hydrothermal synthesis (LTLP). The added advantage of LTHP and LTLP is that the methods are simple, green, provide great added value, are easy to control, and are CO_2_-negative [5]. The same study carried out only recently presented the facile in situ synthesis of multi-layered graphitic carbon nanosheets while using Cu as both template and catalyst via HTC at temperatures below 300 °C. Another advantage presented was the use of natural biomass, with plant leaves of *Pachira aquatica* Aubl (Guiana chestnut) as the carbon source [5].

In another study, Chai et al. (2019) used this method to produce graphene quantum dots (GQDs) from sugarcane bagasse through a process called hot water pre-treatment. This process mostly extracts cellulosic materials without the use of toxic or corrosive media such as organic solvents or alkalis, respectively. In this specific study, low temperatures (170 °C) were used to obtain GQDs with a small size of 2.26 nm. Several sugars were obtained as value-added by-products in addition to the GQDs which can be used in several applications such as solar cells, supercapacitors, the removal and sensing of toxic metal ions in water, and drug delivery, among others [53]. Fluorescent/photoluminescent carbon (nano)dots (CDs) have also been synthesised using a similar method for applications in imaging, sensing, and biolabeling. This was achieved through the hydrothermal treatment of waste wheat straw where it was autoclaved at 250 °C for a short period of 10 h [54]. These methods, however, are still time-consuming even though they are environmentally friendly and cheaper. This then requires the development of more rapid methods that also possess low-cost and green factors to develop carbon materials with the required properties. These methods include those being more specific, microwave-assisted hydrothermal/solid-state carbonisation [55].

### 4.1. Microwave Chemistry for Next-Generation Green Carbon Materials

#### 4.1.1. Hydrothermal Carbonization of Biomass and Other Sustainable Carbon Sources

As mentioned, microwave-assisted hydrothermal carbonisation is a fast method as opposed to conventional HTHP and LTLP. This is attributed to the penetrative nature of microwave radiation where these waves can interact with the internal molecular structure of compatible materials. These materials also include carbon sources such as sugars and biomass/biowastes, which are polar, and thus, microwave-active [55]. These biowastes may encompass celluloses that can be derived from sugar bagasse and other plant wastes [53]. Another usable material is dairy manure as reusable wastes in the production of graphene-like lamellar hydrochars as synthesised by Gao et al. (2018) [56]. The mentioned study reported on the shortened reaction times as required to prepare these hydrochars at the operating temperatures of 240 °C with water as the media. The value-added liquid by-products (light oils) were then extracted using diethyl ether. Another study reported on a facile and green HTC synthesis of a hydrochar through the carbonisation of sugar (glucose). This reaction took place at 200 °C in deionised water over varied reaction times between 5 and 60 min, resulting in largely spherical microparticles after 45 min. These reaction times are a significant improvement from the conventional heating methods which would take many hours to obtain this product. The envisaged applications of such materials include electrodes, adsorbents, and catalyst supports, among others [57].

In another interesting study, Adolfsson and co-workers (2018) presented an HTC carbonisation and post-modification of solid-phase polypropylene in its upcycling as waste material. The method, as described, was one of the few in the literature designed to repurpose waste plastics via microwave-assisted HTC. Furthermore, this method yielded high carbonisation in water while it was incomplete in air. In addition to this, the successful modification with silicon carbide (SiC) was also carried at relatively low temperatures. These achievements are attributed to the in-vessel and self-generated pressure forming favourable subcritical conditions in water [58].

Thermal annealing using microwave irradiation as a reduction approach is another useful method of obtaining fine of carbon nanostructures. This process was reviewed by Lyu et al. (2018) as capable of microwave-generated temperature bursts ca. 2000 °C to occur and in the absence of solvent. This renders the reaction cost-effective and simple to conduct [59]. Jakhar et al. (2020) suggested in their review that the microwave-thermal annealing method provides a superior exfoliation of GO than MWA-HTC [60]. Another study recently presented a two-step preparation of N- and B-doped C-composites prepared from fir bark with ammonium tetraborate (NH_4_B_4_O_7_) as the dopant source. The result of the doping yielded significantly changed surface morphologies and consequently, high surface areas (955 m^2^/g) and enhanced capacitance retention (above 90%) were attained. These properties can be attributed to the favourable synthesis method which resulted in the synergistic improvement of the mentioned properties [61].

In another study, the microwave plasma approach was used to produce large-scale N-doped graphene (N-G) from simple ammonia and ethanol as the precursors. The advantage of this method was that it required no toxic solvents; it is a one-step approach, and as such, it is both cost-effective and environmentally friendly. Furthermore, the reaction takes place at ambient conditions and results in a product possessing high electrical conductivities due to the high purity of the N-G prepared [6]. However, further developments have recently shown that the use of susceptors (supports possessing higher microwave conductivities than the sample material) can significantly enhance the exfoliation as compared to those reported in the above studies [60]. However, at this point in time, the superiority of MWA reduction and annealing of GO seems to largely apply to the exfoliation and inclusion of heteroatoms into the aromatic structure of graphene. Thus, this development does not negate the importance of MWAHTC in terms of the synthesis and modification of GO, rGO, and hydrochars.

#### 4.1.2. Combinatorial MWA-HTC and MWA Exfoliation of GO

Several exfoliation methods were recently developed, including combinatorial approaches. Some of these would be the synthesis of GO using conventional methods followed by MWA exfoliation. To this effect, Voiry et al. (2016) reported 100% yields of highly ordered, single-layer reduced graphene oxide (rGO) from MWA-HTC [48]. Herein, the approach was a simple one in that the GO obtained using the Modified Hummers’ method, was coagulated by a stream of a CaCl_2_ solution into the reaction vessel in the presence of argon. Short bursts (1–2 s) of MW irradiation were then introduced, leading to the reduction of GO, and the formation and annealing of rGO because of the high in situ temperatures (arcing) generated. The intended application of this annealed rGO was towards oxygen evolution reactions. However, the multitude of applications of such advanced carbon materials was pointed out earlier in this review. Thermal annealing using MW irradiation as a reduction approach has been reviewed by Lyu et al. (2018) as resulting in temperature bursts of ca. 2000 °C occurring due to the high energy arcing [49]. Jakhar et al. (2020) suggested in their review that the microwave-thermal annealing method provides superior exfoliation of GO than MWA-HTC [60].

Another study recently presented a two-step preparation of N- and B-doped C-composites prepared from fir bark with ammonium tetraborate as the dopant source. The result of the doping yielded greatly changed surface morphologies and consequently, high surface areas (955 m^2^/g), and enhanced capacitance retention (above 90%). These properties can be credited to the favourable synthesis method which resulted in the synergistic improvement of the mentioned properties [51].

In another study, the microwave plasma approach was used to produce large-scale N-doped graphene (N-G) from simple ammonia and ethanol as the precursors. The advantage of this method was that it required no toxic solvents; it is a one-step approach, and as such, it is both cost-effective and environmentally friendly. Furthermore, the reaction takes place at ambient conditions and results in a product possessing high electrical conductivities due to the high purity of the N-G prepared [6]. However, further developments have recently shown that the use of susceptors (supports possessing higher microwave conductivities than the sample material) can significantly enhance the exfoliation as compared to those reported in the above studies [50]. However, at this point in time, the superiority of MWA reduction and annealing of GO seems to largely apply to the exfoliation and inclusion of heteroatoms into the aromatic structure of graphene. Thus, this development does not negate the importance of MWA-HTC in terms of the synthesis and modification of GO, rGO, and hydrochars.

## 5. Sustainable Carbon Materials in Nanocomposites for Water Treatment and Gas Purification Sustainable Carbon Materials in Membrane Nanotechnology for Water Treatment

Water is a scarce and precious resource the world over, and as such, it is critical to develop methods and materials to harness, protect, and recover it however possible. Nevertheless, climate change and environmental (air/water/soil) pollution remain an unforgiving threat to water sustainability and air quality. Current and conventional methods of addressing these challenges are not sufficient. This is the reason why carbon-based membrane nanotechnology has become much more important in recent years [55,57]. With this knowledge, sustainable, fast, and green methods are needed to develop suitable membranes and membrane materials. Microwave-assisted carbonisation towards the synthesis of these materials are a greatly promising and exciting technology [62].

Other than carbonisation, MWAS was recently used by Ashfaq et al. (2020) to hydrothermally form a polyacrylic acid (PAA) film on a GO-modified commercial TFC membrane. This reaction was also achieved in short periods of 40 s, with the result being an antibiofouling and anti-scaling membrane against *H. aquamarina* (97%) and metal salts (CaCl_2_ and Na_2_SO_4_), respectively. The membrane hydrophilicity and salt rejections were also enhanced with a decrease in permeability due to the grafting which narrowed the pore size [63]. An alternative to this method, however, would be conducting the graft polymerisation of the acrylic acid (AA) to GO in situ with the RO membrane within the MW reactor cell. With the control of parameters such as the monomer concentration and GO loading, a membrane with enhanced bioactive/antibiofouling properties is attainable due to the increased availability of crosslinked GO towards the active layer.

Kovtun et al. (2019) presented a scalable microwave-assisted synthesis (MWAS) method of attaching layered GO to waste polysulfone (PSU) plastics and prepared a mixed matrix membrane (MMM) from the resulting PSU-GO-MW nanocomposite. The PSU-GO synthesis method was a simple (100 °C), fast (45 min), green, and a cost-effective one as no solvents were required; and only an ethanol–water solution was used to wash the product. For comparison, the same product was prepared by conventional oven heating at 200 °C (PSU-GO-OV). Chemical analysis by X-ray photoelectron spectroscopy (XPS) indicated a certain degree of collapse to the molecular structure of the GO nanosheets. The PSU-GO-MW permutation, as such, presented higher Rhodamine B (RhB) and ofloxacin adsorption capacities than both the PSU-GO-OV and pristine PES membranes [64]. This indicates that the microwave chemistry approaches for advanced carbon materials yield MMMs with superior properties as compared to conventional heating.

The important discovery of MWA-HTC has led to critical contributions towards the development of environmentally friendly synthesis methods. The research on the preparation of hydrochars and biochars has resulted in significant interest within the research fraternity of water treatment and membrane nanotechnology. One such study was recently carried out by Hossain et al. (2020) [65]. In this work, rice husk, one of the waste bio-mass materials that continue to generate increasing research interest, was used to prepare a biochar as a sustainable adsorbent which can be included in the fabrication of MMMs for water treatment. The authors synthesised this biochar using a temperature of 70 °C during a reaction period of 20 min at 70 bar working pressure. As such, these reaction parameters improved adsorption capacities and catalytic efficiencies.

### MWAS of Carbon-Based Nanocomposites for Gas Purification

Coal gas coming directly from the bench was historically a noxious chemical soup and it was necessary to eliminate the most deleterious fractions to increase the consistency of the gas, and to avoid exposure to machinery or premises [66]. As such, the elimination of hydrogen sulphide (H_2_S) was assigned the highest priority level at the gas works. To remove this gas, a purifier, housed in a special building and known as an Iron Sponge, was used [67]. The purifier, if the retort-bench itself is not used, was arguably the most significant facility in the gas-works [68]. Originally, purifiers were simple lime-water containers, also referred to as lime cream or milk, where the raw gas from the retort bench was bubbled out to extract hydrogen sulphide. This initial purification method was known as the process of “wet lime”, the lime residue left over from the “toxic wastes” process, gave rise to a substance called “blue billy”, a ferrocyanide contaminant [69].

Recently, researchers have begun to work on redesigning synthesis methods using MW chemistry for many smart materials. Such materials have been found to be quite useful in the fields of power generation, biomaterials, nano-electronics, and nanomedicine, among others. Researchers, scientists, and industrialists are following environmentally benign synthesis routes for the fabrication of desired goods in the age of new science and technology. Of the twelve (12) principles of green chemistry, two major criteria for MWAS are the choice of green solvents and energy efficiency. In many systems, MW heating is seen as a more effective way to manage heating because it is less energy-intensive than traditional methods [70]. The ‘Green Chemistry Monograph of the American Chemical Society’ proposed the use of catalysts or MW irradiation to decrease the energy requirements for the synthesis process [71]. The low-cost processes and versatility of MWAS have also found application in the fabrication of membrane nanotechnology for a wide range of applications.

Membrane nanotechnology is emerging as a critical approach in not only improving the selectivity and affordability of gas sensing, fuel cells, but also gas purification [72]. Therefore, it is critical in the efforts being made towards the mitigation of the effects of air pollution (industrial greenhouse emissions), global warming, and climate change. Carbon-based nanocomposite membranes have incorporated GO, CNTs, and graphene as nanofillers towards gas purification applications such as separating mixtures of CO_2_/CH_4_, and the difficult removal of H_2_S from biogas [73]. Furthermore, they can be used in the separation/capture/adsorption of other gas mixtures of CO_2_/H_2_, H_2_/CH_4_, and N_2_/H_2_, among others, as reviewed by Sazali (2020) [74]. In this work, the author favours the use of carbon membranes over their polymeric counterparts, citing limitations including operating temperatures, poor selectivity in terms of solubility and diffusivity for H_2_ and CO_2_ gases. As such, these challenges influence the design, improvement, and fabrication of such membranes by modifying them with carbon materials.

There are further synthesis methods that have been developed to this effect regarding the green hydrothermal approaches of these carbon materials towards the development of these nanocomposite membranes. One study demonstrated the nanoindentation and the further modification of vertical array C nanotubes (VACNT) via the in situ polymerisation of aniline. The facile method yielded a nanocomposite membrane with a seamless deposition of polyaniline (PANi) onto the said nanotubes, achieving a nanocomposite with enhanced gas transport properties [75]. Figure 4 illustrates the synthesis route and the transmission electron microscopy (TEM) images for the resulting nanocomposite.

## 6. Advances in Solvothermal Synthesis Methods for Advanced Carbon Materials in Gas Purification Technologies

In order to meet the envisaged rising global energy demands and to simultaneously combat the environmental impacts such as global greenhouse gas emissions, it is critical to search for potential energy alternatives [76]. Natural gas is one of such vital components of the world’s supply of energy that has fulfilled the previously mentioned requirements. Natural gas is a cleaner energy source compared to other fossil-based energy sources owing to its low emissions [77]. However, natural gas contains acidic gases (including CO_2_, N_2_, Hg, He, and H_2_S), which may cause equipment corrosion and environmental damage [78]. To date, research has been conducted by Kazmi et al. (2019) amine-based absorption techniques that have been used to remove acidic gases from natural gas to reach regulated concentration limits. However, a tremendous amount of heating is required to regenerate amine-based solvents, which remains a major issue with traditional absorption-based acid gas removal units [79].

Twenty years after the first production of solvothermal reactions, it seems appropriate to review emerging trends in solvothermal reactions,, taking into account their potential and the various economic constraints, identified in the current research activities [80]. Solvothermal reactions have been primarily used in the processing of micro- or nanoparticles of various morphologies over the past twenty (20) years., with research mainly concentrating on determining the potential of solvothermal reactions in material synthesis, and developing new materials that can overcome the disadvantages of conventional treatments, either through fundamental science or by applied research [81]. Researchers have investigated future materials that will be prepared by solvothermal methods, an example of which was illustrated in the work by Pahinkar and Garimella (2018) [82]. The authors of the said work discovered a novel TSA-based gas separation cycle using a microchannel monolith coated along the inner walls of each microchannel with a hollow polymer-adsorbent. By passing impure feed gas through the microchannels, CO_2_ is eliminated from CH_4_, followed by a concurrent flow through the same microchannels of desorbing hot liquid, cooling liquid, and purging gas. Owing to the intimate interaction with the transport fluid and the adsorbent sheet, this configuration is expected to improve the heat and mass transfer to the adsorbent, and thus decrease the total device size because of the flow into the same microchannels of the operating and coupling fluids. Computational models are built in part one of a two-part analysis to study the related fluid dynamics, heat transfer and mass transfer in each phase. Parametric experiments are conducted to determine the optimal geometry of the microchannel and components for adsorbent and heat transfer fluid (HTF). In contrast with bed-based designs, the process is expected to purify up to two orders of magnitude higher gas throughput. A detailed process efficiency chart and the optimisation of energy needs for the process are addressed in the accompanying report.

Zhu et al. (2019) investigated the application of p-conjugated g-C_3_N_4_ (PNa-g-C_3_N_4_) in environmental purification due to its ability to generate reactive-oxygen species (ROS). In their study, the p-conjugated g-C_3_N_4_ (PNa-g-C_3_N_4_) photocatalysts were constructed at the vacancy structure of tri-s-triazine polymer for ROS evolution towards HCHO and NO removal. This occurred through the coordination between the 3p orbitals of Na and the N2p lone electron. The p-conjugated structure increases the ability to absorb visible light and enriches active O_2_ activation sites, thus promoting the directional charge transfer from the N2p of C_3_-N to Na and C. Superior operations, including the evolution of O_2_ (35 mmol/L) and H_2_O_2_ (517 mmol/L) over the photocatalyst PNa-g-C_3_N_4_ have therefore been accomplished. Consequently, PNa-g-C_3_N_4_ photocatalysts display high performance removal efficiency of NO (53% over 6 min contact time) and HCHO (almost 100% over 55 min). These findings may provide a promising future strategy for developing an effective photocatalytic device to produce ROS for environmental purification [83]. Wiheeb et al. (2013) reviewed the technologies available for the removal of hydrogen sulphide (H_2_S) from the gas stream. Biogas, natural gas, and synthesis gas from coal gasification contain H_2_S which is extremely poisonous to humans and corrosive to devices. Prior to utilisation, H_2_S must be separated from fuel gases. The goal of this review was to equate the use of commercial and alkaline impregnated activated carbons to the adsorption of H_2_S. The commercial and alkaline impregnated activated carbons were evaluated for the adsorption of H_2_S at 30 °C and 550 °C by the temperature programme. Much higher H_2_S adsorption was observed with alkaline activated carbons than commercial activated carbon at elevated adsorption temperatures (3–29 times higher depending on the process of modification). In addition, the concentration of H_2_S in the outlet gas after the KOH and Na_2_CO_3_-impregnated activated carbons were treated was less than 30 ppm, which was safe for mechanical and power engine use [84].

Solvothermal carbonisation (STC), as previously mentioned, requires the use of solvents during the carbonisation of carbon precursors. The solvents commonly used in this approach are usually ionic (and toxic) in nature, with some examples being *p*-toluenesulfonic acid, *o*-dichlorobenzene, and 1,2,4,5-tetraaminobenzene, dimethylformamide (DMF), among others [22,85]. In one study, using DMF as the solvent in a solvothermal method, a graphene/metal–organic framework was prepared and it yielded high adsorption capacities for nitrogen gas and benzene [86]. Furthermore, this approach will often require the use of high reaction temperatures, another unwanted reaction parameter as it contributes to high costs. However, there have been several developments to mitigate the effects of or eradicate the use of high temperatures and toxic solvents. These solvents can also be expensive and as such, their negative impacts on the environment need to be reduced. One study in the fairly recent literature by Zhao et al. (2018) used less toxic precursors [87]. The authors utilised millimolar amounts of the organic precursors during a solvothermal synthesis approach tailored to achieve “rigidity derivation”. Rigidity derivation pertains to the use of coplanar organic compounds to enhance conformational rigidity and limit molecular vibrations in the final ACDs. These limited vibrations, respectively, minimise and inhibit non-radiative decay and competitive energy loss. The conformational rigidity was further enhanced by the presence of hydrogen bonds within the molecular structure of HATU. As a result, the nanocrystals produced indicated significantly enhanced quantum yields (29%). The optimised synthesis conditions during the preparation of these ACDs can be seen in Figure 5.

Nevertheless, many studies have reported the use of green solvents such as ethanol a, and the carbon source being biowastes as opposed to organic solvents or other organic molecules [20]. An example of such an investigation was a study conducted by Jin et al. (2017). In this work, the authors synthesised a magnetic nanocomposite material (Fe_3_O_4_/C) from Fe(NO_3_)_3_ 9H_2_O and keratin-rich chicken feathers as beginning biowaste by a green solvothermal method in the presence of ethanol [88].

Han et al. (2020) investigated the reduction of nitric oxide (from anthropogenic pollution), using microbial fuel cells (MFCs) equipped with a gas diffusion cathode. In this study, pure NO was confirmed as the single electron acceptor of gas diffusion cathode (NO-MFC) MFCs. The overall power density and columbic performance were appreciable enhancements in these NO-MFCs relative to MFCs using O_2_ in the air as an electron acceptor (Air−MFCs). The rate of NO elimination was recorded to be 12.33 ± 0.14 mg/L/h and the principal reduction product was N_2_. The dominant route of NO conversion in NO-MFCs, including abiotic electrochemical reduction and microbial denitrification process, was cathode reduction. In the anodic microbial culture, the prevalent genera were observed to change from exoelectrogenic bacteria in air-MFCs to denitrifying bacteria in NO−MFCs and improved the power generation [89].

Due to the problems associated with the release of CO_2_ into the environment, Liu et al. (2017) presented a solvothermal method for the preparation of boron and nitrogen-doped 3D graphene aerogels for carbon capture. This synthesis was carried out at a working temperature of 180 °C during a reaction time of 6 h [90]. Due to the suitability of the synthesis routes and advancements thereof, the carbon materials applied in gas purification and energy generation have gained special characteristics that endow physicochemical stability, regeneration properties, and scalability. As a result, their research and development for industrial applications in the future are highly promising.

## 7. Solvothermal and Hydrothermal Approaches to Synthesis of Carbon Nanodots for Multiple Applications

### 7.1. The Sustainable Synthesis of Carbon Nanodots

This section further discusses CDs and their synthetic routes as initially introduced in the previous sections. This is because of their rising importance over the years due to their versatility. CDs refer to a class of zero-dimension carbon nanoparticles are also known as carbon quantum dots (CQDs), GQDs, carbonised polymer dots (CPDs), or carbogenic nanodots (CNDs) [91,92]. Their carbon cores are usually sp^2^ hybridised with different functional groups such as the amino group, epoxy, ether, carbonyl, aldehyde, hydroxyl, and carboxylic acid on the surface [91,93]. They are quasi-spherical and amorphous to crystalline in nature with sizes less than 10 nm as shown in Figure 5 [91]. These possess unique optical properties (photoluminescence) and differing physicochemical properties [91]. These nanomaterials and their nanocomposites are characterised by facile synthesis methods, water solubility, low cost, biocompatibility, low toxicity, and chemical inertness. Furthermore, CDs are abundant because of the use of inexpensive precursors resulting in sustainable synthesis [94,95]. As a direct result of this, these materials have found widespread applications in anti-counterfeiting, sensing, bioimaging, optoelectronic, energy-related fields, and even wastewater treatment [53,93].

Various synthesis methods have been developed for CD fabrication and these include electrochemical exfoliation of a graphitic source [92]; the incomplete combustion of carbon soot [94]; the carbonisation of polymerised resoles on silica spheres [96]; the thermal oxidation of suitable molecular precursors [97]; and also the dehydration of carbohydrates [98]. Most of these methods often require complex synthesis controls which may cause the adverse degradation of the required CD properties, such as poor crystallinity, the introduction of impurities, extensive post-treatment techniques, as well as complicated purification and separation procedures [99]. Previous reviews have detailed synthesis strategies and different CD applications in bioanalysis, bioimaging, and energy conversion. Such collated knowledge greatly improved our level of understanding and promoted CD development in works that followed [99]. Nevertheless, due to the complicated purification and separation procedures, many challenges are still associated with obtaining CDs with high purity and better physicochemical properties [99].

### 7.2. Solvothermal Synthesis of CDs

In recent years, there have been several studies and reviews on the development of facile synthesis methods that generate enhanced physicochemical properties and surface functionalisation for various applications [95]. From these methods, hydrothermal and solvothermal syntheses have continued to gain popularity [99]. In this regard, the stability of CDs (to maintain structural integrity and minimise leaching) has become of utmost importance. As a result, one of the promising ways is hosting them inside stable structures, such as MOFs and zeolites. A variety of PMs (zeolites, MOFs, mesoporous materials, and other disordered porous nano-carriers) have been used as a host matrix for CDs [91]. Given the diversity and inherent features of porous materials (PMs) and CDs, and the capability of solvothermal and hydrothermal synthesis. A synergistic approach using a mixed synthesis approach of solvothermal synthesis is now possible [91]. In another work, Liu et al. (2017) [100] used a multi-step synthesis approach to synthesise CD-zeolite by carbonising CDs in situ, encapsulating the CDs into zeolite crystals produced by hydrothermal crystallization (Figure 6). This strategy allowed CDs to be tightly confined in the interrupted zeolite framework and form abundant H-bonds. This resulted in the restricted vibration/rotation of the CD functional groups, and thus protected the triplet excitations. The nanocomposite produced indicated unique thermally activated delayed fluorescence (TADF) emissions. In contrast, no TADF emissions were reported by the authors for the pure CDs isolated from the synthesis mother liquid [100].

Pyrolyzing organic solvents or guest molecules confined in MOFs (CDs@MOF nanocomposites) can also be feasibly prepared. Unlike zeolites, the carbonisation of MOFs is usually conducted below 200 °C due to their inferior thermal stability. In addition, the one-step synthesis method used for the synthesis of CDs@zeolites is generally not possible. This is because the precursors of CDs influence the crystallization of MOFs and the synthesis temperatures required for the preparation of MOFs is also much lower than the carbonisation temperature of CDs [91,101]. This means that the most suitable synthesis routes for carbon-modified MOFs are the LTLP approaches as discussed in Section 4.1.

These methods involve the use of inexpensive precursors, simple synthesis processes, environmentally friendly approaches, and non-toxic routes [95]. Since most of the prepared CD-based materials already possess an abundance of hydrophilic functional groups, no additional modification treatment is required to impart hydrophilic properties and reactivity to the CDs [95]. An example of this was presented by Zhu et al. [102] where they developed a simple microwave-assisted method to synthesise new types of CDs. In that work, a carbohydrate was used as a carbon source and polyethylene glycol (PEG200) was employed as both a solvent and coating agent. The reaction gradually changed from colourless to a dark brown solution when under 500 W microwave irradiation for a duration ranging 2~10 min. The product was diluted with water to attain fluorescent CDs. The obtained particle size and the quantum yields of fluorescent CDs were observed to depend on the reaction time [102].

Recent studies indicate that short external heat pulses can contribute to the chemical oxidation and carbonisation of organics and convert them into CDs. Microwaves with frequencies ranging from 300 MHz to 300 GHz provide sufficient energy to break the chemical bonds in the raw materials [95]. The microwave-assisted hydrothermal/solvothermal synthesis provides uniform heating, thus effectively reducing the reaction time so that the particle size distribution of CDs is small. Additionally, MWAS does not require a hydrothermal reactor for the sealing and reaction of the organic precursor at high pressure, high temperatures, and long reaction times compared with conventional synthesis [103].

### 7.3. Hydrothermal Synthesis of CD Composites

Although CDs have various advantages, they are also prone to problems associated with the aggregation of pristine CDs in solid state. This often leads to property change such as luminescence quenching, surface defects, and decreasing surface areas [91]. To mitigate these unwanted phenomena, and to further optimise their optical and electrical properties, various solid supports such as polymer matrices, inorganic salts, and porous materials (PMs) have been used to support CDs [104,105]. To prepare such nanocomposites, solvothermal and hydrothermal synthesis techniques are at the forefront of establishing various modified CDs. Typically, for CD nanocomposites, two main synthesis approaches have been proposed, i.e., a one-step method and a two-step method [93].

The one-step method means that the simultaneous generation of the CDs into the supporting matrix is carried out in the same reaction. The two-step method means that the CDs are first prepared using the top–down/bottom–up route and then embedded in the host matrices via chemical or physical methods [93]. A good example of the latter was demonstrated by Ming et al. [99] where a novel photocatalyst (TiO_2_/CD) was prepared by combining CDs with TiO_2_ through an easy hydrothermal method. The obtained TiO_2_/CDs exhibited excellent visible-light photocatalytic activity. The CDs were initially prepared through the electrochemical exfoliation of a graphitic source as follows: drop-wise addition (1 mL/min) of Ti-[OCH(CH_3_)_2_]_4_ (1 mL, dissolved in absolute alcohol with a ratio of 1:19) into CDs solution (55 mL, 0.1 mg/mL). After the continuous stirring of the mixture for 4 h, a colloidal solution of TiO_2_/CDs nanohybrids then formed. The resulting solution was sealed into a Teflon-lined autoclave, followed by hydrothermal treatment at 180 °C for 48 h. The obtained gray solid TiO_2_/CDs was washed with water and ethanol, and dried in a vacuum oven at 80 °C.

The direct in situ incorporation of CDs into the host matrix is somewhat more difficult with very few studies reported in the literature [93]. This is mainly due to difficulties in controlling the simultaneous generation of CDs in one system and the requirement of a strong driving force to co-assemble the guest CDs into the host matrix [106]. Nevertheless, Zhang et al. (2019) recently synthesised luminescent CDs@zeolite and CDs@AlPO nanocomposites through direct in situ incorporation of luminescent CDs into the host matrix using the solvothermal approach. In this method, triethylamine was utilised as the template and triethylene glycol as the solvent to form uniformly embedded CDs (≈3.7 nm) in AlPO-5 crystals as shown in Figure 7 [107].

The successful one-step synthesis of CDs@zeolite composites was credited to the facile solvothermal synthesis method. The end-result was the highly dispersed CDs due to the well-confined nanospaces and high stability of zeolites. In this case, the organic species (i.e., templates and solvents) used in the synthesis of zeolites provide the source materials for CDs and the simultaneous formation of CDs and zeolites can further be achieved by varying certain synthesis conditions [93]. The use of one-step reaction processes not only simplifies the preparation procedure compared with the multistep preparation method, since the as-prepared CDs also indicates enhanced fluorescence, crystallinity, and uniform size. Different CDs with varying properties can be achieved by choosing different solvents as the carbon source [108].

Thus far, a variety of PMs (zeolites, MOFs, mesoporous materials, and other disordered porous nano-carriers) have been used as the host matrix for CDs [91]. Considering the diversity and inherent features of PMs and CDs, and the capability of solvothermal and hydrothermal synthesis, a synergistic approach using a mixed synthesis approach of solvothermal synthesis is therefore possible [91].

## 8. An Overview of the Various Methods, Advantages, Disadvantages, Properties, and Applications

The methods as discussed in this review have their own advantages and disadvantages which impart certain properties that render them either applicable or inapplicable for the appropriate applications. As these have now been reviewed, their summary in terms of the precursors, reaction conditions, and the resulting carbon material is given in Table 1. From this table, it becomes easier to select a method based on the specific requirements regarding the properties and the application of the carbon material or nanocomposite to be prepared. For example, one could prepare a magnetic Fe_3_O_4_/MMC from solvothermal synthesis for the effective adsorption of environmental pollutants as per Chen et al. (2020) [109]. However, excessively high temperatures are required to obtain the desired structures to attain the necessary adsorption capacities. However, other studies have innovated to avert the use of high temperatures while reporting effective carbonisation following solvothermal synthesis. One such study is also illustrated in the table where Zhao et al. (2018) prepared amphiphilic carbon dots (ACDs) at a low carbonization temperature of 150 °C during a shorter reaction period of 8 h [87]. Low reaction temperatures and short reaction times assume further benefits such as reduced operational costs while obtaining materials with high quantum yields, as reported in the latter study. An alternative to the latter method, a plasma-enhanced CVD method (PECVD), was developed by Dissanayake et al. (2016) to fabricate energy and sensing devices [43]. The authors were able to perform this synthesis of low temperatures as opposed to other approaches such as solvothermal and CVD methods in the literature. The only disadvantage of this approach is the number of steps involved in the fabrication, in as much as it may be scalable. Within the requirements of lowering energy demand during the synthesis of these carbon materials and nanocomposites, advances have also contributed to developments of sol–gel synthesis. For example, Escamilla et al. (2020) presented significantly reduced calcination times for the preparation mesoporous carbon-supported TiO_2_ (TiO_2_/MC, to 2 h at 400 °C). As such, this method not only benefited from lower reaction temperatures, but decreased reaction times and the resulting TiO_2_/MC possessed enhanced textural properties [48]. On the other hand, conventional arc-discharge is widely reported for its high-power consumption as the reactions are conducted under high temperatures. However, the non-thermal arc-discharge (NTAC) method reported by Wang and co-workers averted the utilization of heat [41]. This is the significant advantage of such a method for the reasons already discussed. Furthermore, the graphene nanoflakes were produced in only 30 min reaction time, achieving high surface areas with a wide potential window for the application of this material.

The high temperatures and pressures required by conventional hydrothermal synthesis also place the method at a high disadvantage, and as such, LTHP and LTLP methods were developed, including HTC. However, conventional HTC is still associated with extended reaction times, even though the temperatures are significantly reduced. Such was the case reported by Yuan and colleagues (2015) [54]. However, the use of sustainable biowastes as feedstocks for this HTC approach makes it favourable for the preparation of CDs. Furthermore, these CDs possess a uniform size, spherical morphology, and photoluminescent properties. These properties render them suitable for utilization in applications such as sensing, bio-imaging, the imaging of inorganic ions, and fluorescent inks. Due to the long reaction times as required by conventional hydrothermal and solvothermal approaches, MWAS has gained considerable attention. This is because it also minimises the environmental impacts of the reactions, improves safety, and results in cleaner products, and is scalable. One study by Khavlyuk et al. (2021) compared the effect of solvent polarity on the resulting CDs prepared via conventional STC and MWA-STC [49]. The only obvious advantage MWAS showed over STC was the highly rapid reaction times. As a result, there were differences in the final morphologies of the CDs, and these were attributed to the fact that in STC, the reactions are governed by temperatures which promote carbonization. In MWA-STC, the reactions are dictated by the pressure which results in the formation of molecular surface moieties, and hence the formation of onion-ring-like structures in this study. In addition to MWA-STC, MWA-HTC methods have also been explored in the recent past because of the outstanding advantages they possess. A study by Chae et al. (2017) reported the MWA-HTC synthesis of CDs from succinic acid (SA) and tris(2-aminoethyl)amine (TAEA) was used as a template. Similarly to other MWAS approaches, this method was exceedingly fast, taking between 3–5 min to complete. This reaction is faster than many methods discussed in this review [50].

Sustainable feedstocks such as biowastes, waste plastics, and sugars have also been carbonised using MWA-HTC to produce biochars, hydrochars, and other carbon materials. One such study was conducted by Adolfsson and co-workers to produce a SiC/PP nanocomposite. The disadvantage of this method was the requirement of harsh chemicals that aided the carbonization of the waste PP. However, there were several advantages to this approach such as the upcycling of waste plastics, which are an environmental nuisance. Furthermore, the reactions were fast and taking only between 20 and 80 min to complete, and the depositing the SiC onto PP easily took place under dry MW conditions [58]. Mixed-matrix membranes (MMMs) have also been prepared from similar approaches under dry conditions as presented by Kovtun et al. (2019) [64]. This reaction took place without the use of any chemicals and was compared with its counterpart where the GO was attached to the polysulfone (PSU) scraps at elevated temperatures. All the properties investigated for the respective resulting materials were reported to be enhanced for the membranes prepared via MWAS.

## 9. Outlook for Synthesis Methods and Considerations for the Future

With the demand for carbon materials steadily rising, it is evident that green and sustainable synthesis methods are required to produce cost-effective and stable carbon materials for multidimensional applications. The current advancements in hydrothermal, solvothermal, and other synthesis methods for carbon materials indicate that there are many research opportunities. In terms of solvothermal methods, generally, there is a need to drastically reduce the use of toxic solvents to safeguard the environment and human health. An example of this would be the use of alcohol–water mixtures, during synthesis where studies still lack. There are highly commendable efforts currently being made by researchers towards the use of biowastes, waste plastics, and waste solvents to convert them into biochars, hydrochars, CDs, other carbon materials, and value-added by-products. One of the carbon sources currently gaining research interest is waste cotton from textile items since a significant amount of these is being discarded. Another important factor is energy demand that needs to be drastically lowered for HTHP processes. There are some methods that exist in the literature that utilise low temperatures and shortened thermal treatment times. One such study was conducted by per Escamilla et al. (2020) where calcination times were reduced to just 2 h. However, further research into such reactions need to be conducted [48]. This can be achieved through combinatorial approaches such as MWA-sol–gel-synthesis. This method is mainly used for the preparation of metal oxides applied in the energy field. Nevertheless, the synthesis methods for other applications (as noted in this review) still require attention. Sol–gel mixing, drying, and grinding, followed by dry MWA-arcing or MWA-plasma are also routes also worth exploring. Such a route could significantly shorten synthesis lead times. In the case of non-conductive materials being synthesised in this manner, the use of susceptors can enhance conductivity and increase conversion rates and yields. Significant improvement in the suggestions made in this section could mean that the large-scale production of advanced carbon materials can be realised in the future.

## 10. Conclusions

This review has highlighted and discussed the synthesis methods as employed in the preparation of advanced carbon materials and their resulting nanocomposites for various applications such as energy, sensing, gas purification, water treatment, and environmental remediation. The synthesis methods as disseminated, include solvothermal, hydrothermal, and to an extent, a few others that have shown promise from 2015 to date. The different permutations and developments pertaining to these methods were also discussed in terms of their precursors, reaction conditions, morphologies, the resulting properties, and the final or proposed/suitable applications. In terms of the precursors, the importance of feedstocks was also highlighted, as this dictates the final properties and performance of the final material. Furthermore, carbon feedstocks determine whether the synthesis method is sustainable or not. As such, biomass, biowastes, waste plastics, and solvents were discussed. The use of solvents and harsh chemicals was also discussed and discredited because of human health and environmental threats. Additionally, these chemicals have been shown in the literature to be very difficult and expensive to remove, and this introduces further cleaning costs. As such, the use of these chemicals in these synthesis methods was noted as a significant disadvantage and deterrent to future advancements and large-scale production. Further disadvantages were noted to be contributed to by high energy demands and excessively long reactions which increase production costs and thereby resulting in high product costs. Advantageous reactions have been shown to mainly include STC, HTC, MWA-HTC, and MWA-STC. In terms of both STC and HTC, this was attributed to the lower reaction temperatures, shorter reaction times, reduced use of solvents and harsh chemicals. For MWA-HTC and MWA-STC, the benefits far outweigh those listed for their conventional counterparts as the reactions are complete in significantly higher speeds, the products are cleaner, the morphologies and properties are notably improved. For example, the solvent/acid/water-free MWA-exfoliation of GO, which produces high-quality rGO, with benefits highly improved compared to other reported methods. Further benefits from MWAS then result in enhanced catalysis, adsorption, sensitivity, among other desired material characteristics. As such, this review highly recommends further studies and investigations into the application of MWAS as a method of advanced carbon materials synthesis. These advanced carbon materials can then be applied in fuel cells, photovoltaics, the degradation/adsorption of in-/organic environmental insults; and in-membrane nanotechnology for gas separation/storage/adsorption; and water purification.

## Figures and Tables

**Figure 1 materials-14-05094-f001:**
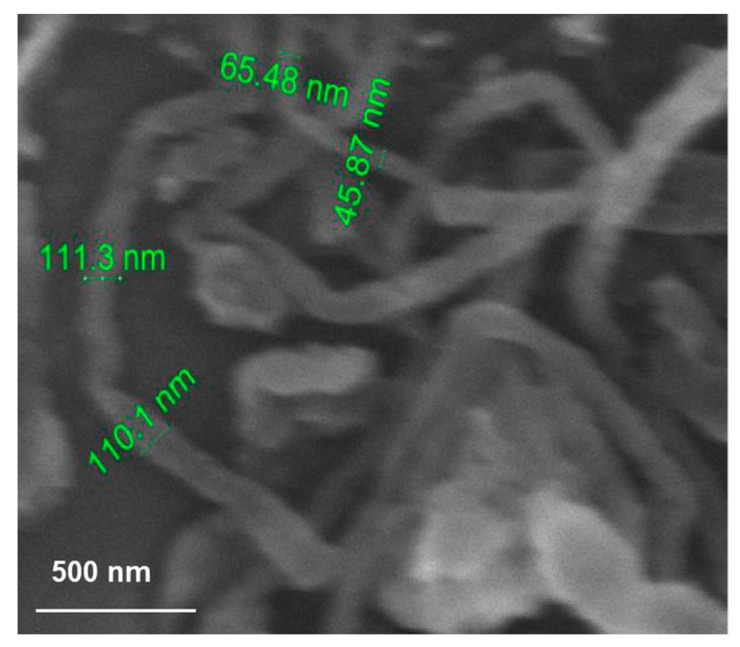
A SEM micrograph of MWCNTs synthesised by the arc-discharge method. Reproduced from Sharma et al. (2015) with permission from Elsevier B.V. © 2021 [39].

**Figure 2 materials-14-05094-f002:**
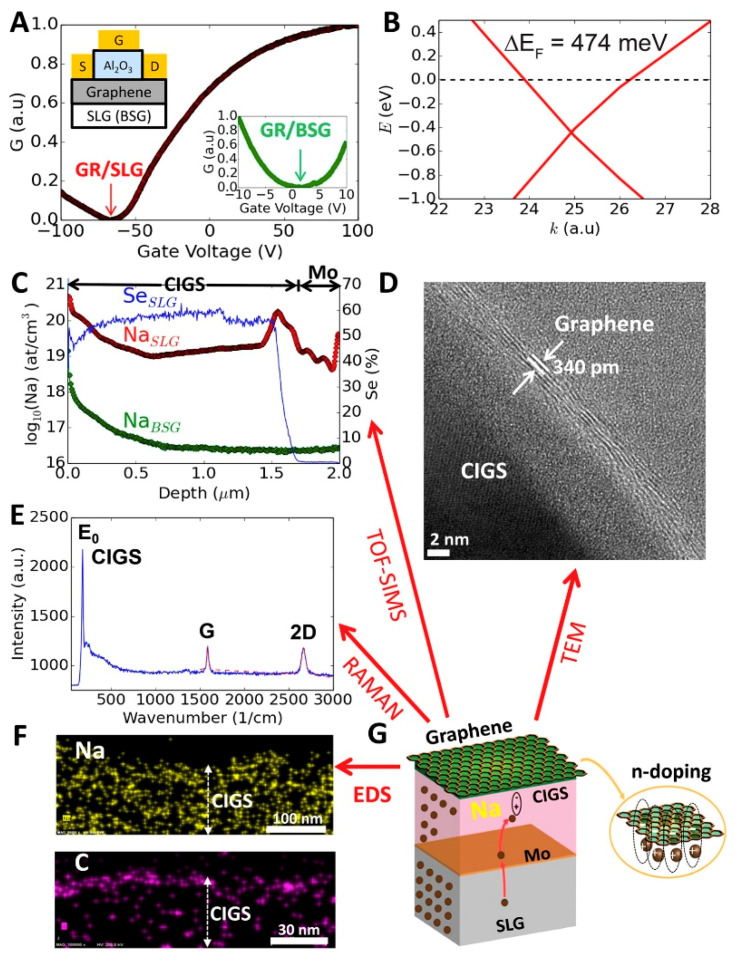
(**A**) Normalised conductance (G) vs. gate-voltage (V_G_) of G (GR)/soda-lime glass (SLG) and GR/borosilicate glass (BSG) (inset) measured in FET configuration; (**B**) DFT curve showing G–Na n-doping interactions; (**C**) Na and Se depth-profiles in CIGS/Mo/SLG and CIGS/Mo/BSG from TOF-SIMS; (**D**) cross-sectional HR-TEM image of GR/CIGS/Mo/SLG; (**E**) Raman spectra of GR/CIGS/Mo/SLG; (**F**) EDS mapping of GR/CIGS/Mo/SLG showing Na (yellow, top) and C (purple, bottom); and (**G**) Schematic of G n-doping mechanism on CIGS. Reproduced from Dissanayake et al. (2016) [43].

**Figure 3 materials-14-05094-f003:**
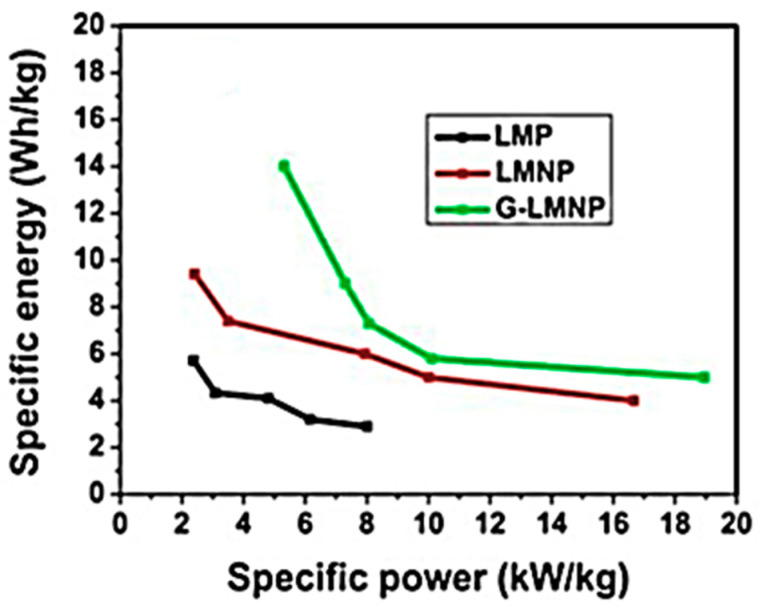
The Ragone plots of AC // LMP, AC // LMNP and AC // GLMNP lithium–ion capacitors. Adapted from Hlongwa et al. (2020) with permission from John Wiley & Sons Inc. © 2021 [45].

**Figure 4 materials-14-05094-f004:**
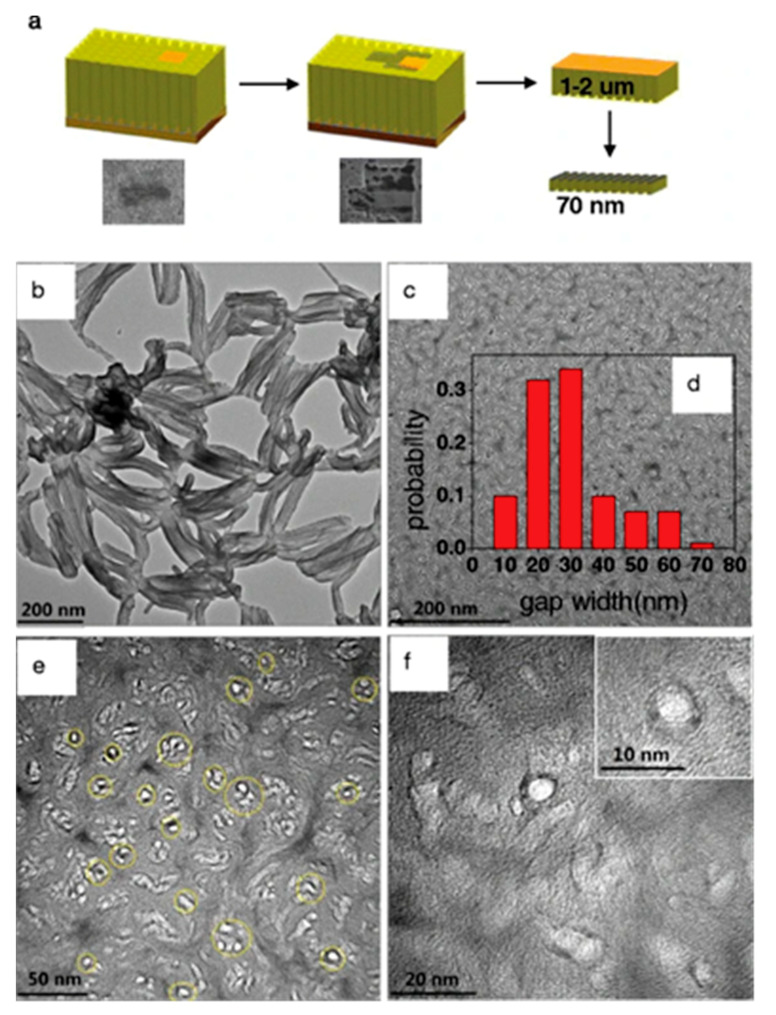
(**a**) Focused ion beam (FIB) sample preparation (nanoindentation) and TEM micrographs; (**b**) cross-sectional TEM images of VACNT; (**c**,**e**,**f**) TEM images of the VACNTs/PANi nanocomposite membrane cross-section at different resolutions; and (**d**) histogram of the gap width distribution between the nanotubes (insert on (**c**)). Reproduced from Ding et al. (2015) [75] under the Creative Commons Licence BY 4.0. Springer Nature. Copyright © 2021, Ding et al.

**Figure 5 materials-14-05094-f005:**
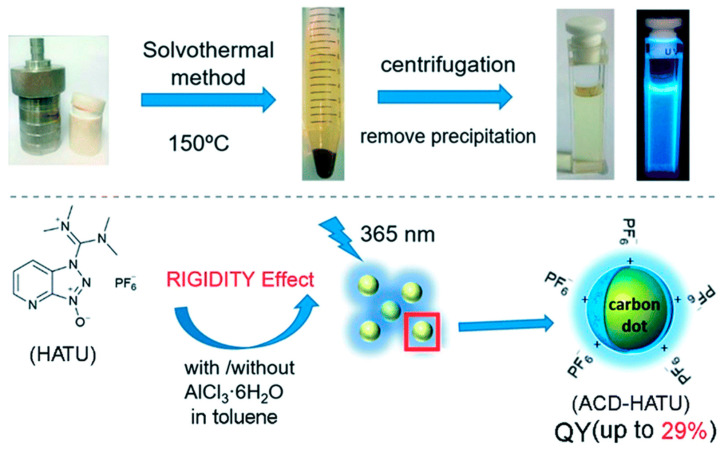
The synthesis of ACD-HATU. Reproduced from Zhao et al. (2018) [87] under the Creative Commons 3.0 license © 2021 Royal Society of Chemistry.

**Figure 6 materials-14-05094-f006:**
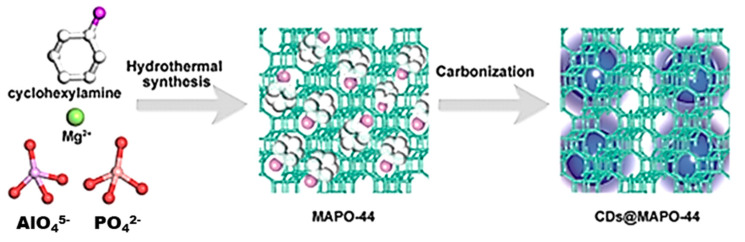
The multi-step synthesis route for the preparation of CDs@MAPO-44 where the final product shows CDs confined in MAPO-44 (CHA) via the carbonisation of the organic template. Adapted from Zhang et al. (2020) [91] with permission from John Wiley & Sons, Inc. © 2021.

**Figure 7 materials-14-05094-f007:**
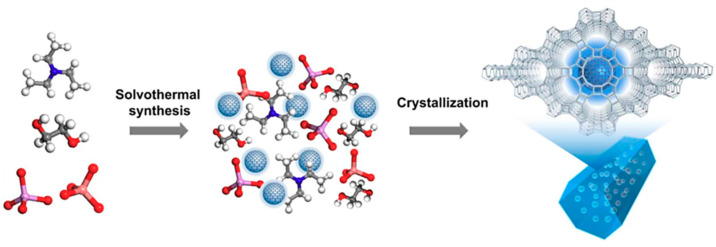
Schematic illustration of one-step synthesis: a) CDs@AlPO-5 composite via solvothermal synthesis. Adapted from Zhang et al. (2020) [91] with permission from John Wiley & Sons, Inc. © 2021.

**Table 1 materials-14-05094-t001:** The various synthesis methods, advantages, disadvantages, properties, and the applications of carbon materials.

Method	Precursors	Conditions	Product	Properties	Advantages (A)/Disadvantages (D)	Application/s
MWA-HTC [50]	Succinic acid (SA), tris(2-aminoethyl)amine (TAEA)	700 W, 3–5 min in water	CQDs	Spherical, photoluminescent	D: Household MW–caution against chemical fume exposure. A: Easy, fast. Small, uniform QD size, high sensitivity, selectivity	Drug delivery, bioimaging, photo/electrocatalysis
Freeze-drying and thermal treatment [110]	GO, wood, polyvinyl acrylate (PVA) 1400	Various followed by carbonisation (400 °C, 30 min; 300 °C, 2 h)	Carbonised wood cell chamber (CWCC)-rGO@PVA	Flexibility, 3D porous structure, layered, high surface areas, capacitive, adsorptive	A: Simple method; stable nanocomposite; enhanced textural properties; high energy densities; high specific capacitance; good electrochemical performance D: Multiple treatment steps	Supercapacitors, batteries, sensors
PECVD [43]	SiO_2_, graphene	Various temperatures (140–160 °C)	Graphene/soda lime glass.	Semiconducting	D: complicated process-many steps/conditions. A: Scalable, cheap.	Electronics, batteries, sensors, photovoltaics, etc.
Solvothermal synthesis [110]	Pluronic P123 template, Fe salt, chitosan	Calcination, 700–900 °C, 12 h	Fe_3_O_4_/C (magnetic mesoporous carbon (M MC)	Spherical particles, magnetic, adsorptive.	D: High calcination temperatures. A: Efficient dye adsorption. A: Highly adsorptive material	Adsorbent for environmental remediation
MWA-HTC [58]	SiC, waste PP	210–250 °C, 20–80 min	Amorphous C and SiC/C	Semi-crystalline particles	D: Use of harsh chemicals. A: Upcycling; easy and fast methods	Potential applications in agriculture, catalysis, adsorption, electrochemistry, etc.
MWA-HTC [111]	Coffee grounds,	180 °C, 40 bar in water	QDs	Particles 6 nm in size, spherical, crystalline	A: Scalable, fast; small and uniform particle size	Removal of dyes in water using nanoporous graphene membranes
Sol–gel with thermal treatment [48]	MC, TiO_2_, Ti(OC_3_H_7_)_4_,	Calcination at 400 °C, 2 h	TiO_2_/MC	Crystalline; homogeneous dispersion of TiO_2_ on MC; photocatalytic	A: Improved speeds; lower temperatures; enhanced textural properties	Hydrogen production
Non-thermal plasma arc-discharge [41]	CH_4_	Ar/H_2_ plasma, ambient temperature, 20–200 kPa, 200 W, 30 min	Graphene nanoflakes	High surface areas, high crystallinity, small particle size; high thermal stability	A: Rapid, low energy consumption; ambient operating temperatures,	A wide window of potential applications
MWA-HTC vs. HTS [56]	Dairy manure	240 °C, 4 h in water	Biochar	Graphene-like lamellar structure; microspheres	A: Improved structures over HTC; high yields, green method; value-added by-products	Possible applications in supercapacitors and adsorption of dyes in water
HTC [54]	Wheat straw	250 °C, 10 h in water,	CDs	Amorphous structure, nanospheres, photoluminescence	A: Water soluble CDs, D: Long reactions	Sensing, bio-imaging, imaging of inorganic ions; fluorescent inks, etc.
Solvothermal [87]	1-[bis(dimethylamino)methylene]- 1H-1,2,3-triazolo [4,5-b]pyridinium 3-oxide hexafluorophosphate (HATU), 1-benzylimidazole, 1 mL diisopropylamine, AlCl_3_·6H_2_O	150 °C, 8 h	Amphiphilic CDs (ACDs)	Small (2–5 nm), monodisperse, spherical, crystalline nanoparticles; photoluminescence	A: One-step reaction; facile method; high quantum yields (QY); low toxicity	Bioimaging, optoelectronics
MWA-LTH vs. oven heating [64]	Polysulfone (PSU) scraps, GO	70 °C, 45 min, dry conditions	PSU-GO-OV (oven); PSU-GO-MW (MW)	Layered GO on PSU	A: Low temperatures, green, fast, upcycling, stable, adsorptive, reusable, recovery of adsorbent. PSU-GO-MW—better properties than PSU-GO-OV.	Adsorption of dyes from water
Solvothermal [86]	GO, ZnNO_3_·6H_2_O, 1,4-benzene-dicarboxylate,	120 °C, 25 h in DMF, CHCl_3_	GO/MOF	Microporous to mesoporous, high surface areas, adsorptive	A: High adsorption capacities. D: Long reactions; use of toxic chemicals	Adsorption of volatile organic compounds (VOCs)
Solvothermal [88]	Waste chicken feathers (CF), Fe(NO_3_)_3_·9H_2_O	200 °C, 30 h in EtOh	Fe_3_O_4_/CF	Magnetic, adsorptive	A: Simple, green reactions, D: Long reactions	Potential adsorbent for environmental pollutants from water
MWA dry reduction and annealing [112]	GO, CaCl_2_,	1. Mild reduction: 300 °C, 1 h under Ar 2. Annealing: 1–2 s under Ar	rGO	Single layer rGO	A: Highly ordered structures;	Oxygen evolution reactions
STC [113]	Wood (lignin-free), NiCl_2_·9H_2_O, TAEA, Bis-cyclic carbonate	Polymerisation, and vitrimization (25 °C, DMSO)	Carbonised wood/Ni/Nis	Structurally improved wood; flower-like morphology of Ni/Nis	A: Sustainable biomass; good dispersion of the nanoNi/NiS; high power densities; Enhanced conductivity; cycling stability; high energy densities; self-healing; etc.	CO_2_ reduction to Ch_3_OH; supercapacitors, etc.
Solvothermal vs. MWA-STC [49]	Phloroglucinol, DMF, EDA, FA	S: 160 °C, 6 h; M: 700 W, 6 min	N-CDs	Amorphous, carbonised, to onion-like structures depending on solvent.	(S) D: Time-consuming. (M) A: highly rapid synthesis speeds	Potential is catalysis, optoelectronics, etc.

## Data Availability

Data sharing not applicable. No new research data were created or analysed in this study.

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
