# Peer review of "Sustainable Hydrothermal and Solvothermal Synthesis of Advanced Carbon Materials in Multidimensional Applications: A Review"

_materials, 2021, doi:10.3390/ma14175094_

Round 1

Reviewer 1 Report

Comments: In this review, the advanced carbon materials prepared by hydrothermal and solvothermal methods and their applications in different fields are summarized. But before it can be accepted, there are still some minor problems that need to be improved.

  1. In recent years, some advanced carbon materials prepared by other methods and their applications should also be mentioned, so as to attract more readers' interest and attention. Such as Journal of Materials Chemistry A 8 (2020) 10898-10908; Cellulose 28 (2021) 1455-1468; Chemical Engineering Journal 418 (2021) 129518; Cellulose 28 (2021):3733-3743; Journal of Power Sources. 471 (2020) 228448; Industrial Crops & Products 167C (2021) 113545.
  2. What is the author's outlook for the development of related fields in the future? The author had better be able to elaborate from the existing problems and development trends.
  3. There are still some small format problems in the references. For example, the names of periodicals are abbreviated in some places and full names in some places. Please check carefully and revise to keep consistent.

Author Response

Thank you for reviewing the manuscript. Your comments assisted in improving its quality. Because the language needed to be improved, the manuscript was sent for language editing by a first language English speaker. Their corrections are highlighted in yellow on both our responses and the corrected manuscript accompanying this document.

Thank you.

Reviewer 2 Report

This paper shows a review of hydrothermal, solvothermal, and other advanced carbon materials synthetic methods. There are some issues that need to address:

- I believe this review would benefit from a table that could compare and provide an overview of the discussed approaches. The table should include the advantages and limitations of each approach.

- The language of the paper needs to be improved, as such, it is really difficult to read...

- Introduction is written simply, most recent research and innovation advanced carbon materials synthetic performances should be reviewed to show the gap of knowledge. An introduction should be extended with recently research papers. The introduction should be rewritten to show the highlights and novelty of the work.

Line 80 and 88: “, this section of the book chapter aims to address the recent developments……   other carbon precursors shall also be looked at in this book chapter.”, Book chapter?!

- quality figures 4, 6, and 7 are poor.

- section of drawbacks and future could be increased quality of the manuscript.

- Similar reviews have been published recently. It is recommended to add a statement to clearly separate the current work from these similar references and also define the review period (e.g. last five years). Also, prepare statistical data (such as the number of documents, document per country) about you used references by created databank such as Scopus, Google scholar, and web of science.

- Should be provided a comprehensive 2-3 tables between all of these fields till now used.

- The conclusion section is very short and general. Must be rewritten and summarized.

Author Response

Thank you for reviewing our manuscript. Your comments helped improve the language, structure, science, narrative, and discussion, among other important things about the manuscript. The  language, as corrected by native English speaker, appears highlighted in yellow throughout the responses to your comments and the accompanying corrected manuscript.

Round 2

Reviewer 2 Report

The comments of my first report have been addressed by the authors.